# Effects of current on wind waves in strong winds

*Naohisa Takagaki[1], Naoya Suzuki[2], Yuliya Troitskaya[3], Chiaki Tanaka[2],*

*Alexander Kandaurov[3], Maxim Vdovin[3]*

[1] (Corresponding Author) Department of Mechanical Engineering, University of Hyogo, Shosha 2167, Himeji Hyogo, 671-2280 Japan, *E-mail: takagaki@eng.u-hyogo.ac.jp*
Tel/Fax: +81-79-267-47834

[2] Faculty of Science and Engineering, Kindai University, 3-4-1, Kowakae Higashiosaka Osaka, 577-8502 Japan.

[3] Department of Geophysical Research, Institute of Applied Physics, the Russian Academy of Sciences, 46 Ul'yanov Street, Nizhny Novgorod, 603-950, Russia.

**Keywords:** wind waves, current, Doppler shift

**Abstract**

It is important to investigate the effects of current on wind waves, called the Doppler shift, both at normal and extreme high wind speeds. Three different types of wind-wave tanks along with a fan and pump are used to demonstrate wind waves and currents in laboratories at Kyoto University, Japan, Kindai University, Japan, and the Institute of Applied Physics, Russian Academy of Sciences, Russia. Profiles of the wind and current velocities and the water-level fluctuation are measured. The wave frequency, wavelength, and phase velocity of the significant waves are calculated, and the water velocities at the water surface and in the bulk of the water are also estimated by the current distribution. The study investigated 27 cases with the measurements of winds, waves, and currents, at wind speeds ranging from 7 to 67 m s$^{-1}$. At normal wind speeds under 30 m s$^{-1}$, wave frequency, wavelength, and phase velocity depend on wind speed and fetch. The effect of the Doppler shift is confirmed at normal wind speeds, i.e., the significant waves are accelerated by the surface current. The phase velocity can be represented as the sum of the surface current and artificial phase velocity, which is estimated by the dispersion relation of the deep-water waves. At extreme high wind speeds, over 30 m s$^{-1}$, a similar Doppler shift is observed as under the conditions of normal wind speeds. This suggests that the Doppler shift is an adequate model for representing the acceleration of wind

waves by current, not only for the wind waves at normal wind speeds but also for those with intensive breaking at extreme high wind speeds. A weakly nonlinear model of surface waves at a shear flow is developed. It is shown that it describes well the dispersion properties of not only small-amplitude waves but also strongly nonlinear and even breaking waves, typical for extreme wind conditions (over 30 m s$^{-1}$).

## 1. Introduction

The oceans flow constantly, depending on the rotation of the Earth, tides, topography, and wind shear. High-speed continuous ocean flows are called currents. Although the mean surface velocity of the ocean is approximately 0.1 m s$^{-1}$, the maximum current surface velocity is more than 1 m s$^{-1}$ (e.g., Kawabe, 1988; Kelly et al., 2001). The interaction between the current and wind waves generated by wind shear have been investigated in several studies. The acceleration effects of the current on wind waves, called the Doppler shift; the effects of the current on the momentum and heat transfer across the sea surface; and the modeling of waves and currents in the Gulf Stream have been the subject of experimental and numerical investigations (e.g., Dawe and Thompson, 2006; Kara et al., 2007; Fan et al., 2009; Shi and Bourassa, 2019). Thus, wind waves follow the dispersion relationship and Doppler shift effect at normal wind speeds. However, these studies were performed at normal wind speeds only, and few studies have been conducted at extreme high wind speeds, for which the threshold velocity is 30 – 35 m s$^{-1}$, representing the regime shift of the air-sea momentum, heat, and mass transport (Powell et al., 2003; Donelan et al., 2004; Takagaki et al., 2012, 2016; Troitskaya et al., 2012, 2020; Iwano et al., 2013; Krall and Jähne, 2014; Komori et al., 2018; Krall et al., 2019). At such extremely high wind speeds, the water surface is intensively broken by the strong wind shear, along with the foam layer, dispersed droplets, and entrained bubbles (e.g. Donelan et al., 2004; Troitskaya et al., 2012, 2017, 2018a, 2018b; Takagaki et al., 2012, 2016; Holthuijsen et al., 2012). It is unclear if the properties of wind waves and the surface foam layer at extremely high wind speeds are similar to those at normal wind speeds. Furthermore, in a hurricane, the local ocean flows may be unusually strong, change rapidly, and strongly affect wind waves. However, the effects of the current on wind waves have not yet been clarified.

Therefore, the purpose of this study is to investigate the effects of the current on wind waves in strong winds through the application of three different types of wind-wave tanks, along with a pump.

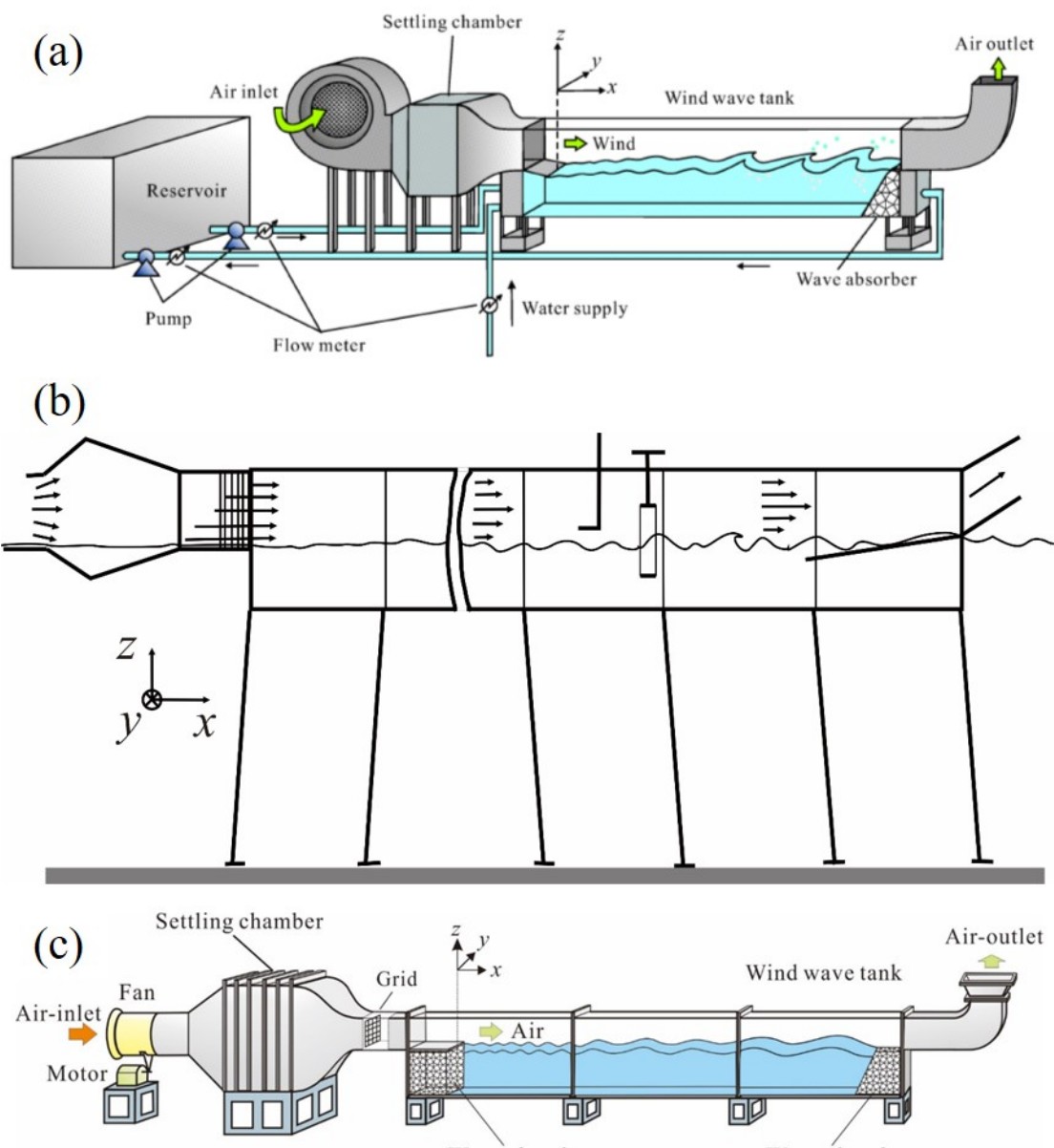

**Figure 1.** Schematics of wind-wave tanks. (a) High-speed wind-wave tank of Kyoto University. (b) Typhoon simulator of IAP RAS. (c) Wind-wave tank of Kindai University.

## 2. Experiment

### 2.1. Equipment and measurement methods

Wind-wave tanks at Kyoto University, Japan and the Institute of Applied Physics, Russian Academy of Sciences (IAP RAS) were used in the experiments (Figs. 1a, 1b). For the tank at Kyoto University, the glass test section was 15 m long, 0.8 m wide, and 1.6 m high. The water depth $D$ was set at 0.8 m. For the tank at IAP RAS, the test section in

the air side was 15 m long, 0.4 m wide, and 0.4 m high. The water depth $D$ was set at 1.5
m. The wind was set to blow over the filtered tap water in these tanks, generating wind
waves. The wind speeds ranged from 4.7 to 43 m s$^{-1}$ and from 8.5 to 21 m s$^{-1}$ in the tanks
at Kyoto and IAP RAS, respectively. Measurements of the wind speeds, water-level
fluctuation, and current were carried out 6.5 m downstream from the edge ($x = 0$ m) in
both the Kyoto and IAP RAS tanks. Here, the $x$, $y$, and $z$ coordinates are referred to as the
streamwise, spanwise, and vertical directions, respectively, with the origin located at the
center of the edge of the entrance plate. Additionally, the fetch ($x$) is defined as the
distance between the origin and measurement point ($x = 6.5$ m).

In Kyoto, a laser Doppler anemometer (Dantec Dynamics LDA) and phase Doppler

anemometer (Dantec Dynamics PDA) were used to measure the wind velocity fluctuation.
A high-power multi-line argon-ion (Ar$^+$) laser (Lexel model 95-7; laser wavelengths of
488.0 and 514.5 nm) with a power of 3 W was used. The Ar$^+$ laser beam was shot through
the sidewall (glass) of the tank. Scattered particles with a diameter of approximately 1 μm
were produced by a fog generator (Dantec Dynamics F2010 Plus) and were fed into the
air flow over the waves (see Takagaki et al. (2012) and Komori et al. (2018) for details).
The wind speed values ($U_{10}$) at a height of 10 m heigh above the ocean and the friction
velocity ($u^*$) were estimated by the eddy correlation method, by which the mean velocity
($U$) and the Reynolds stress ($-uv$) in air were measured. The $u^*$ was estimated by an eddy
correlation method as $u^* = (-\langle uv \rangle)^{1/2}$, because the shear stress at the interface ($\tau$) was
defined by $\tau = \rho u^{*2} = \rho C_D U_{10}^2$. The value of $(-\langle uv \rangle)^{1/2}$ was estimated by extrapolating the
measured values of the Reynolds stress to the mean surface of $z = 0$ m. The $U_{10}$ was
estimated by the log-law: $U_{10} - U_{min} = u^*/\kappa \ln(z_{10}/z_{min})$, where $U_{min}$ is the air velocity
nearest the water surface ($z_{min}$) and $z_{10}$ is 10 m. Moreover, the drag coefficient $C_D$ was
estimated by $C_D = (u^*/U_{10})^2$.

Water level fluctuations were measured using resistance-type wave gauges (Kenek

CHT4-HR60BNC) in Kyoto. The resistance wire was placed into the water, and the
electrical resistance at the instantaneous water level was recorded at 500 Hz for 600 s
using a digital recorder (Sony EX-UT10). The energy of the wind waves ($E$) was
estimated by integrating the spectrum of the water-level fluctuations over the frequency
($f$). The values of the wavelength ($L_S$) and phase velocity ($C_S$) were estimated using the
cross-spectrum method (e.g., Takagaki et al., 2017) (see the detail in Appendix). The
current was measured using the same LDA system.

In IAP RAS, a hot-wire anemometer (E+E Electrinik EE75) was used to measure the

representative mean wind velocity at $x = 0.5$ m and $z = 0.2$ m. The three wind velocities
($U_{10}$, $u^*$, $U_\infty$) at $x = 6.5$ m were taken from Troitskaya et al. (2012) by a Pitot tube. Here,
$U_\infty$ is the freestream wind speed. The $u^*$ was estimated by a profile method considering
the profiles in the constant flux layer and the wake region:

$$U_\infty - U(z) = u^*\left(-\frac{1}{\kappa}ln(z/\delta) + \alpha\right); \ z/\delta < 0.15, \qquad (1)$$


$$U_\infty - U(z) = \beta u^*(1 - z/\delta)^2; \ z/\delta > 0.15, \qquad (2)$$


respectively. Here, $\delta$ is the boundary layer thickness, and $\alpha$ and $\beta$ are the constant values
that depend on flow fields and are calibrated at low wind speeds without the dispersed
droplets. At extremely high wind speeds, measuring the profile in the constant flux layer
(Eq. 1) is difficult because of the large waves; thus, using $\beta$ measured at low wind speeds,
$u^*$ is estimated by Eq. (2). The value of $U_{10}$ is estimated by Eq. (1) at $z_{10} = 10$ m with
measured $\alpha$ at normal wind speeds. The value of $C_D$ is estimated by $C_D = (u^*/U_{10})^2$.
Although the measurement methods for $u^*$, $U_{10}$, and $C_D$ in IAP RAS and Kyoto are
different, the values approximately correspond to each other (see Troitskaya et al. (2012)
and Takagaki et al. (2012)).
The water-level fluctuations were measured using three handmade capacitive-type
wave gauges in IAP RAS. Three wires formed a triangle with 25 mm on a side
($x$-directional distance between wires $\Delta x$ is 21.7 mm). The wires were placed in the water,
and the output voltages at the instantaneous water level were recorded at 200 Hz for 5400
s using a digital recorder through an AD converter (L-Card E14-140). The values ($E$, $f_m$,
$H_S$, $T_S$, $C_S$, and $L_S$) were estimated by the same mannar as in Kyoto tank. The current was
measured through acoustic Doppler velocimetry (Nortec AS) at $x = 6.5$ m and $z = -10$,
$-30$, $-50$, $-100$, $-150$, $-220$, and $-380$ mm (see Troitskaya et al. (2012) for details).

**2.2. Artificial current experiments at Kindai University**
Additional experiments were performed using a wind-wave tank at Kindai
University with a glass test section 6.5 m long, 0.3 m wide, and 0.8 m high (Fig. 1c) (e.g.
Takagaki et al., 2020). The water depth $D$ was set at 0.49 m. A Pitot tube (Okano Works,
LK-0) and differential manometers (Delta Ohm HD402T) were used to measure the mean
wind velocity. The values of $u^*$, $U_{10}$, and $C_D$ (Cases 21-27) were estimated using $U_\infty$ by
the empirical curve by Iwano et al. (2013), which was proposed by the eddy correlation
method used in Kyoto (see section 2.1).
The water level fluctuations were measured using resistance-type wave gauges
(Kenek CHT4-HR60BNC). To measure $L_S$ and $C_S$, another wave gauge was fixed
downstream at $\Delta x = 0.02$ m, where $\Delta x$ is the interval between the two wave gauges. The
values ($E$, $f_m$, $H_S$, $T_S$, $C_S$, and $L_S$) were estimated by the same mannar as in Kyoto tank.
The current was then measured through electromagnetic velocimetry (Kenek LP3100)
with a probe (Kenek LPT-200-09PS) at $x = 4.0$ m. The probe sensing station was 22 mm
long with a diameter of 9 mm. The measurements were performed at $z = -15$ to $-315$ mm
at 30 mm intervals. The sampling frequency was 8 Hz, and the sampling time was 180 s.

**3. Results and discussion**
**3.1. Waves and current**
Figure 2 shows the vertical distributions of the streamwise water velocity. The
water velocities in the three different wind-wave tanks at Kyoto University, Kindai
University, and IAP RAS are separately shown in each subfigure. In Fig. 2a, the bulk
velocity of water $U_{BULK}$ shows negative values ($U_{BULK} = -0.16$ to $-0.01$ m s$^{-1}$) at Kyoto
University, which is generated as the counterflow against the Stokes drift at the wavy
water surface. In Fig. 2b, the bulk velocity of water demonstrates positive values ($U_{BULK}$
$= 0.019$ to $0.044$ m/s) at IAP RAS, because the wind-wave flume is submerged; thus, the
Stokes drift on the wavy water surface does not provide the counterflow for the bulk
water, unlike in the closed tank at Kyoto University. From Fig. 2c, it is clear that the bulk
velocities of the water vary in each case at Kindai University with the use of the pump.
Furthermore, the water bulk velocities change from negative to positive ($U_{BULK} = -0.13$
to $-0.17$ m s$^{-1}$). The bulk velocities of water were defined as the mean velocity with $z =$
$-0.4$ to $-0.25$ m (see dotted lines in Fig. 2), and the velocities are listed in Table 1.
Experiments were performed under 27 different conditions, with the bulk velocity of
water provided in the three different wind-wave tanks. The surface velocities of water,
$U_{SURF}$, also varied in the three tanks with respect to wind speed (see Fig. 2). The $U_{SURF}$
values were estimated by the linear extrapolation lines (dashed lines) as the water velocity
at the surface ($z = 0$ m) shown in Fig. 2, and the velocities are listed in Table 1.
Figure 3 shows the wind-velocity dependency of the wave frequency $f_m$,
wavelength $L_S$, phase velocity $C_S$, surface velocity of water $U_{SURF}$, and bulk velocity of
water $U_{BULK}$. From Figs. 3a–3c, it is clear that both the Kyoto and IAP RAS data
demonstrate that the wind waves develop with wind shear. Although $f_m$ in both cases
correspond to each other, $L_S$ and $C_S$ in IAP RAS are different from those in Kyoto. The
disagreement might be caused by the difference in the wind-wave development or
Doppler effect; this is discussed below. From Figs. 3d and 3e, $U_{SURF}$ and $U_{BULK}$ increase
with an increase in $U_{10}$ in IAP RAS. However, in Kyoto, $U_{SURF}$ increases, but $U_{BULK}$
decreases with an increase in $U_{10}$. Moreover, $U_{SURF}$ in IAP RAS corresponds to $U_{SURF}$ in
Kyoto. This is because the Stokes drift generated by the wind waves, rather than the
current, is significant. For the Kindai data, although $f_m$, $U_{SURF}$, and $U_{BULK}$ vary, $L_S$ and $C_S$
are concentrated at single points at $L_S = 0.1$ m and $C_S = 0.4$ m s$^{-1}$, respectively.

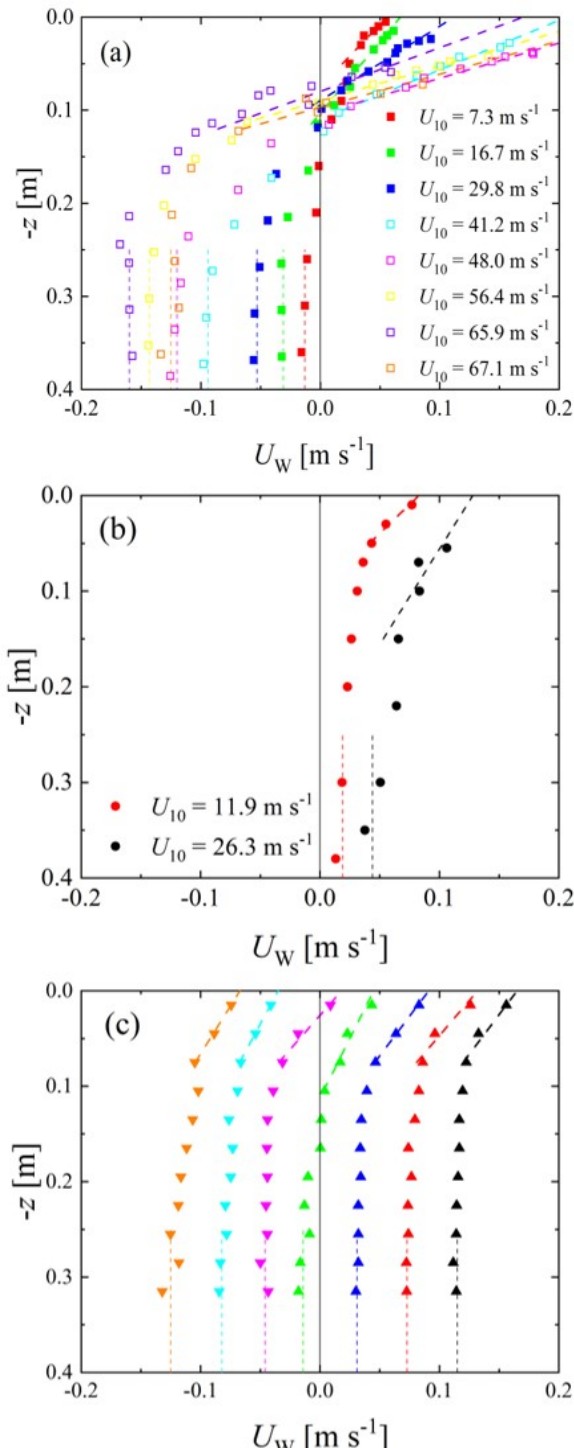


**Figure 2.** Vertical distributions of water-flow velocity; (a) Kyoto University, (b) IAP RAS, and (c)

Kindai University. In (c), plots indicate cases 21–27 starting from right. Dotted and dashed lines

indicate the lines used to estimate $U_{BULK}$ and $U_{SURF}$, respectively. Open symbols show the

high-wind-speed cases.

**TABLE 1.** Wind and wind-wave properties. $F$: fetch; $N_{PUMP}$: pump inverter frequency; $U_\infty$:
freestream wind speed; $u^*$: friction velocity of air; $U_{10}$: wind speed at 10 m above the sea surface;
$U_{SURF}$: surface flow velocity of water; $U_{BULK}$: bulk flow velocity of water; $C_D$: drag coefficient; $H_S$:
significant wave height; $T_S$: significant wave period; $E$: wave energy; $f_m$: significant frequency; $C_S$:
phase velocity; $L_S$: significant wave length; $C_{S\text{-theor-l}}$: phase velocity predicted by theoretical linear
model; $C_{S\text{-theor-nl}}$: phase velocity predicted by theoretical nonlinear model. The values of $u^*$, $U_{10}$, and
$C_D$ in Kindai were estimated using the empirical curves by Iwano et al. (2013) from $U_\infty$. Superscripts †
and †† indicate the artificial following and opposing flows, respectively.

| Case | Facility | $F$ [m] | $N_{pump}$ [Hz] | $U_\infty$ [m s⁻¹] | $u^*$ [m s⁻¹] | $U_{10}$ [m s⁻¹] | $U_{SURF}$ [m s⁻¹] | $U_{BULK}$ [m s⁻¹] | $C_D$ [×10⁻³] | $H_s$ [m] | $T_s$ [m] | $E^{0.5}$ [m] | $f_m$ [Hz] | $C_s$ [m s⁻¹] | $L_s$ [m] | $C_{s\text{-theor-l}}$ [m s⁻¹] | $C_{s\text{-theor-nl}}$ [m s⁻¹] |
|---|---|---|---|---|---|---|---|---|---|---|---|---|---|---|---|---|---|
| 1 | Kyoto | 6.5 | - | 4.7 | 0.24 | 7.3 | 0.056 | -0.01 | 1.1 | 0.0035 | 0.15 | 0.00092 | 6.63 | 0.40 | 0.06 | 0.369 | 0.374 |
| 2 | Kyoto | 6.5 | - | 7.2 | 0.43 | 11.5 | - | - | 1.4 | 0.0131 | 0.25 | 0.00353 | 3.95 | 0.59 | 0.16 | - | - |
| 3 | Kyoto | 6.5 | - | 10.3 | 0.67 | 16.7 | 0.067 | -0.031 | 1.6 | 0.0231 | 0.32 | 0.00624 | 3.03 | 0.69 | 0.23 | 0.658 | 0.690 |
| 4 | Kyoto | 6.5 | - | 12.6 | 0.89 | 21.5 | - | - | 1.7 | 0.0357 | 0.39 | 0.00968 | 2.59 | 0.92 | 0.38 | - | - |
| 5 | Kyoto | 6.5 | - | 16.3 | 1.49 | 29.8 | 0.112 | -0.053 | 2.5 | 0.0584 | 0.50 | 0.01570 | 2.01 | 1.09 | 0.52 | 0.972 | 1.044 |
| 6 | Kyoto | 6.5 | - | 18.8 | 1.70 | 33.6 | - | - | 2.5 | 0.0626 | 0.52 | 0.01691 | 1.89 | 1.18 | 0.60 | - | - |
| 7 | Kyoto | 6.5 | - | 22.2 | 2.08 | 41.2 | 0.206 | -0.094 | 2.6 | 0.0631 | 0.53 | 0.01735 | 1.86 | 1.35 | 0.74 | 1.188 | 1.258 |
| 8 | Kyoto | 6.5 | - | 24.8 | - | - | - | - | - | 0.0668 | 0.55 | 0.01866 | 1.76 | 1.41 | 0.79 | - | - |
| 9 | Kyoto | 6.5 | - | 28.5 | 2.36 | 48.0 | 0.273 | -0.120 | 2.4 | 0.0727 | 0.58 | 0.02058 | 1.68 | 1.54 | 0.93 | 1.325 | 1.424 |
| 10 | Kyoto | 6.5 | - | 31.1 | - | - | - | - | - | 0.0807 | 0.62 | 0.02309 | 1.58 | 1.60 | 1.07 | - | - |
| 11 | Kyoto | 6.5 | - | 34.8 | 2.69 | 56.4 | 0.241 | -0.143 | 2.3 | 0.0944 | 0.68 | 0.02715 | 1.44 | 1.64 | 1.10 | 1.379 | 1.550 |
| 12 | Kyoto | 6.5 | - | 37.1 | 2.89 | 57.7 | - | - | 2.5 | 0.1043 | 0.73 | 0.03027 | 1.37 | 1.76 | 1.31 | - | - |
| 13 | Kyoto | 6.5 | - | 39.6 | 3.38 | 65.9 | 0.170 | -0.160 | 2.6 | 0.1214 | 0.80 | 0.03553 | 1.20 | 1.84 | 1.51 | 1.531 | 1.694 |
| 14 | Kyoto | 6.5 | - | 43.3 | 3.31 | 67.1 | 0.272 | -0.125 | 2.4 | 0.1609 | 0.93 | 0.04766 | 1.08 | 2.01 | 1.92 | 1.743 | 2.149 |
| 15 | IAP RAS | 6.5 | - | 8.5 | 0.40 | 11.9 | 0.083 | 0.019 | 1.1 | 0.0214 | 0.31 | 0.0056 | 3.14 | 0.78 | 0.25 | 0.690 | 0.715 |
| 16 | IAP RAS | 6.5 | - | 11.0 | 0.60 | 16.7 | - | - | 1.3 | 0.0305 | 0.36 | 0.0081 | 2.84 | 0.89 | 0.32 | - | - |
| 17 | IAP RAS | 6.5 | - | 13.5 | 0.90 | 21.9 | - | - | 1.7 | 0.0455 | 0.43 | 0.0121 | 2.41 | 1.07 | 0.45 | - | - |
| 18 | IAP RAS | 6.5 | - | 16.3 | 1.15 | 26.3 | 0.128 | 0.044 | 1.9 | 0.0790 | 0.50 | 0.0161 | 1.95 | 1.27 | 0.65 | 1.111 | 1.190 |
| 19 | IAP RAS | 6.5 | - | 18.9 | 1.50 | 32.5 | - | - | 2.1 | 0.0690 | 0.54 | 0.0246 | 1.85 | 1.37 | 0.74 | - | - |
| 20 | IAP RAS | 6.5 | - | 21.2 | 1.70 | 36.9 | - | - | 2.1 | 0.0847 | 0.60 | 0.0305 | 1.61 | 1.61 | 1.00 | - | - |
| 21 | Kindai | 4.0 | 15† | 5.8 | 0.28 | 7.9 | 0.165 | 0.115 | 1.2 | 0.0044 | 0.14 | 0.0012 | 6.92 | 0.43 | 0.06 | 0.484 | 0.492 |
| 22 | Kindai | 4.0 | 10† | 5.8 | 0.28 | 7.9 | 0.132 | 0.073 | 1.2 | 0.0050 | 0.16 | 0.0014 | 6.10 | 0.43 | 0.07 | 0.501 | 0.510 |
| 23 | Kindai | 4.0 | 5† | 5.8 | 0.28 | 7.9 | 0.091 | 0.031 | 1.2 | 0.0049 | 0.16 | 0.0014 | 6.16 | 0.38 | 0.06 | 0.410 | 0.420 |
| 24 | Kindai | 4.0 | 0 | 5.8 | 0.28 | 7.9 | 0.045 | -0.014 | 1.2 | 0.0054 | 0.19 | 0.0014 | 5.47 | 0.38 | 0.07 | 0.382 | 0.393 |
| 25 | Kindai | 4.0 | 5†† | 5.8 | 0.28 | 7.9 | 0.018 | -0.046 | 1.2 | 0.0076 | 0.23 | 0.0021 | 4.25 | 0.36 | 0.08 | 0.384 | 0.400 |
| 26 | Kindai | 4.0 | 10†† | 5.8 | 0.28 | 7.9 | -0.035 | -0.082 | 1.2 | 0.0098 | 0.27 | 0.0027 | 3.64 | 0.35 | 0.10 | 0.355 | 0.375 |
| 27 | Kindai | 4.0 | 15†† | 5.8 | 0.28 | 7.9 | -0.067 | -0.125 | 1.2 | 0.0125 | 0.34 | 0.0035 | 2.94 | 0.38 | 0.13 | 0.381 | 0.402 |

This shows that the intensity and direction of the current do not significantly affect $L_S$ and
$C_S$ but do affect $f_m$ and $U_{SURF}$. Thus, this implies that the present artificial current changes
the water flow dramatically but does not affect the development of the wind waves.
Figure 4 shows the dispersion relation and demonstrates that the Kindai data
points depend on the variation in the water velocity of the artificial current. The plots for
the Kyoto University and IAP RAS cases at normal wind speeds (solid symbols) are
concentrated above the solid curve, showing the dispersion relation of the deep-water

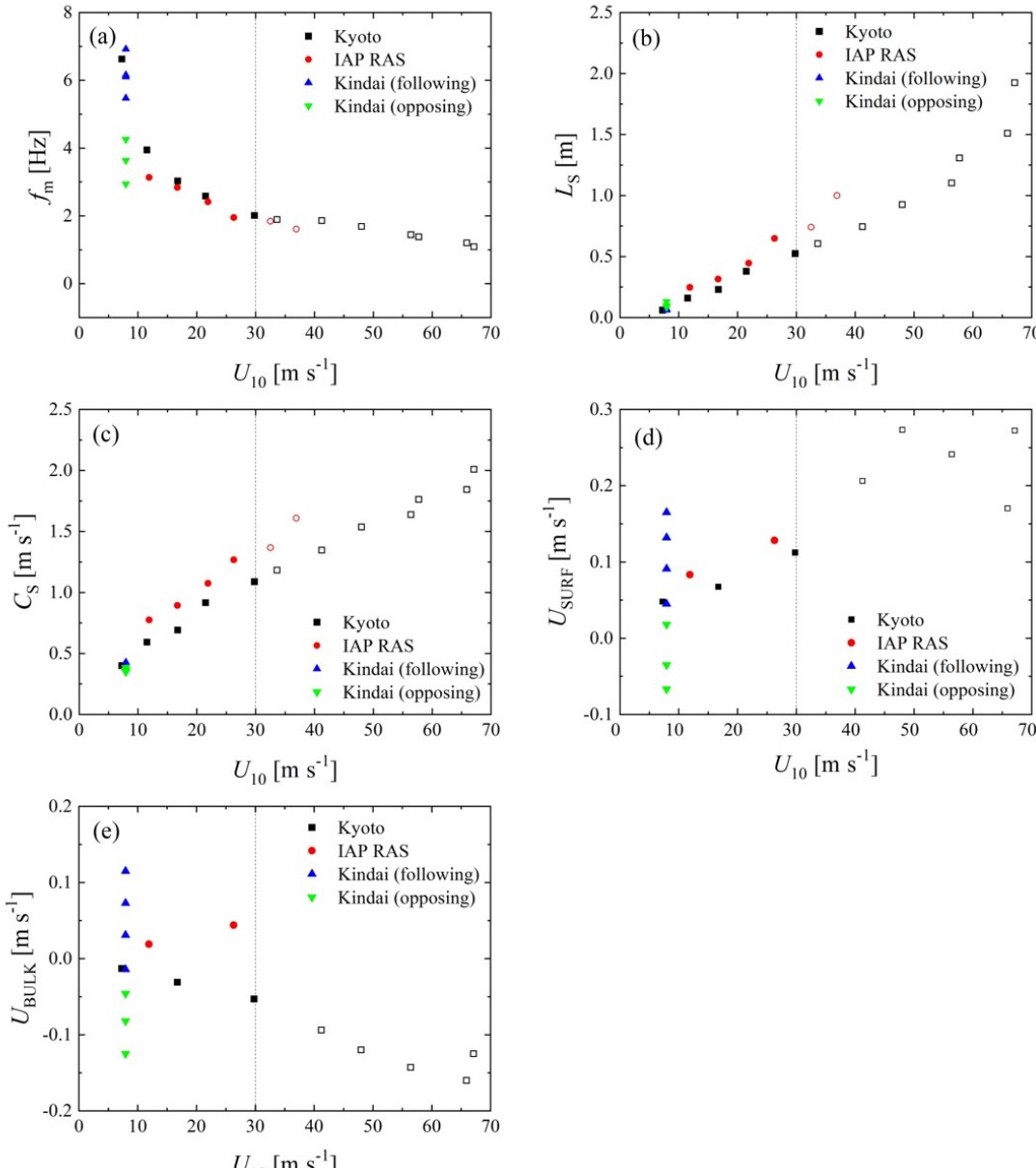

**Figure 3.** Relationships between $U_{10}$ and (a) significant frequency $f_{\mathrm{m}}$, (b) significant wave length $L_{\mathrm{S}}$, (c) phase velocity $C_{\mathrm{S}}$, (d) surface velocity of water $U_{\mathrm{SURF}}$, and (e) bulk velocity of water $U_{\mathrm{BULK}}$. Open symbols show the high-wind-speed cases.

waves ($\omega^2 = gk$). Meanwhile, the plots for extreme high wind speeds (open symbols) are also concentrated above the solid curve. This implies that the wind waves, along with the intensive breaking at extreme high wind speeds, are dependent on the Doppler shift. To investigate the phase velocity trend, Fig. 5 shows the ratio of the measured phase velocity

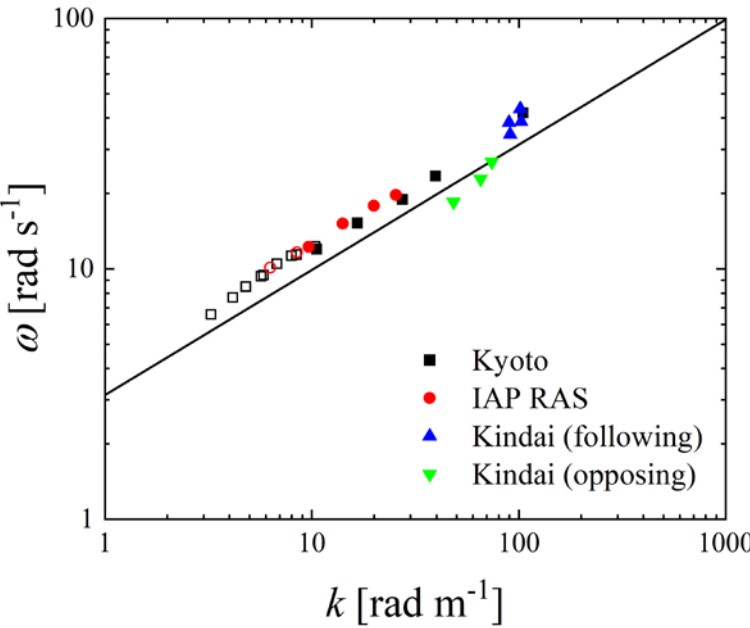


**Figure 4.** Dispersion relation between angular frequency $\omega$ and wave number $k$. Open symbols show
the high-wind-speed cases. Curve shows the dispersion relation of the deep-water waves ($\omega^2 = gk$).

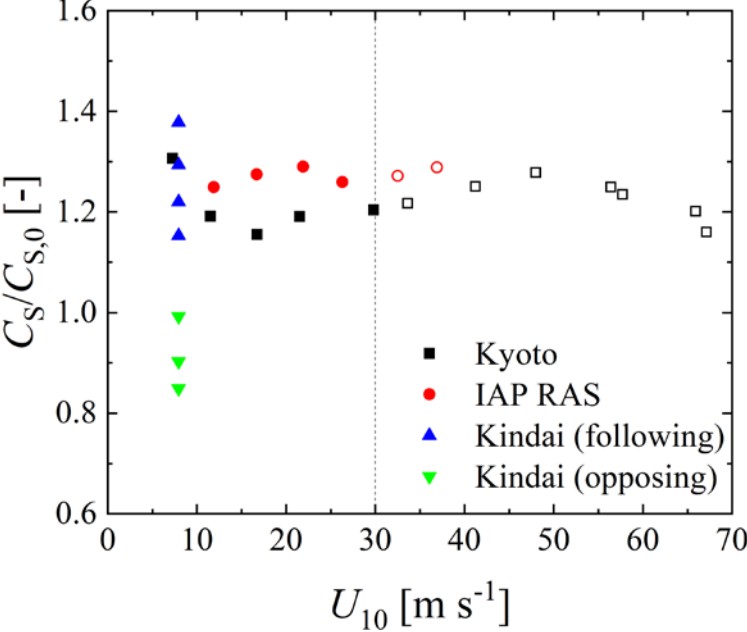


**Figure 5.** Relationship between the freestream wind speed and phase velocity $C_S$. The $C_S$ is
normalized by phase velocity $C_{S,0}$ without the Doppler effect, estimated by the dispersion relation of
the deep-water waves ($C_{S,0} = (gL_S/2\pi)^{1/2}$). Open symbols show the high-wind-speed cases.
$C_S$ to the phase velocity $C_{S,0}$ estimated by the dispersion relation of the deep-water waves
($C_{S,0} = (gL_S/2\pi)^{1/2}$) against the wind velocity. From the figure, the ratios at the normal
wind speeds assume a constant value (~1.21 in Kyoto or ~1.27 in IAP RAS). Moreover,
the ratios at the extreme high wind speeds take similar values of 1.23 and 1.28 for Kyoto
or IAP RAS, respectively. This implies that the phase velocities at extreme high wind
speeds are accelerated by the current just as those at normal wind speeds. However, the
Kindai values are scattered and increase in the following cases and decrease in the
opposing cases. It is clear that the artificial current accelerates (or decelerates) the phase
velocity.
To interpret the relationship among the measured phase velocity $C_S$, first phase
velocity $C_{S,0}$ estimated by the dispersion relation, and water velocity, two types of phase
velocity were evaluated: the sum of $C_{S,0}$ and surface velocity of water $U_{SURF}$ and the sum
of $C_{S,0}$ and bulk velocity of water $U_{BULK}$. Figure 6 shows the relationship between $C_S$ and
(a) $C_{S,0} + U_{SURF}$, and (b) $C_{S,0} + U_{BULK}$. In Fig. 6a, we can see that the Doppler shift is
confirmed at the normal wind speeds, i.e., the significant waves are accelerated by the
surface flow, and the real phase velocity can be represented as the sum of the velocity of
the surface flow and the virtual phase velocity, which is estimated by the dispersion
relation of the deep-water waves. At extreme high wind speeds over 30 m s$^{-1}$, a similar
Doppler shift is observed as under the conditions of normal wind speeds, as seen in Fig.
6a. Meanwhile, in Fig. 6b, although $C_S$ corresponds to $C_{S,0} + U_{BULK}$ at low phase
velocities, $C_S$ assumes values larger than $C_{S,0} + U_{BULK}$ at high phase velocities. This
suggests that the Doppler shift is an adequate model for representing the acceleration of
the wind waves by the current, not only for the wind waves at normal wind speeds but
also for those with intensive breaking at extreme high wind speeds. Moreover, the
Doppler shift of wind waves occurs due to a very thin surface flow, as the correlation
between $C_S$ and $C_{S,0} + U_{SURF}$ is higher than the correlation between $C_S$ and $C_{S,0} + U_{BULK}$.
**3.2. The theoretical model of waves at the shear flow**
The parameters of the observed Doppler shift can be explained more precisely
within the theoretical model of the capillary-gravity waves at the surface of the water
flows with the velocity profiles prescribed by the experimental data, which are plotted in
Fig. 2a–c. Because the dominant wind wave propagates along the wave and water flows,
we will consider the 2D-wave model in the 2D flow. This flow is described by the system
of 2D Euler equations:

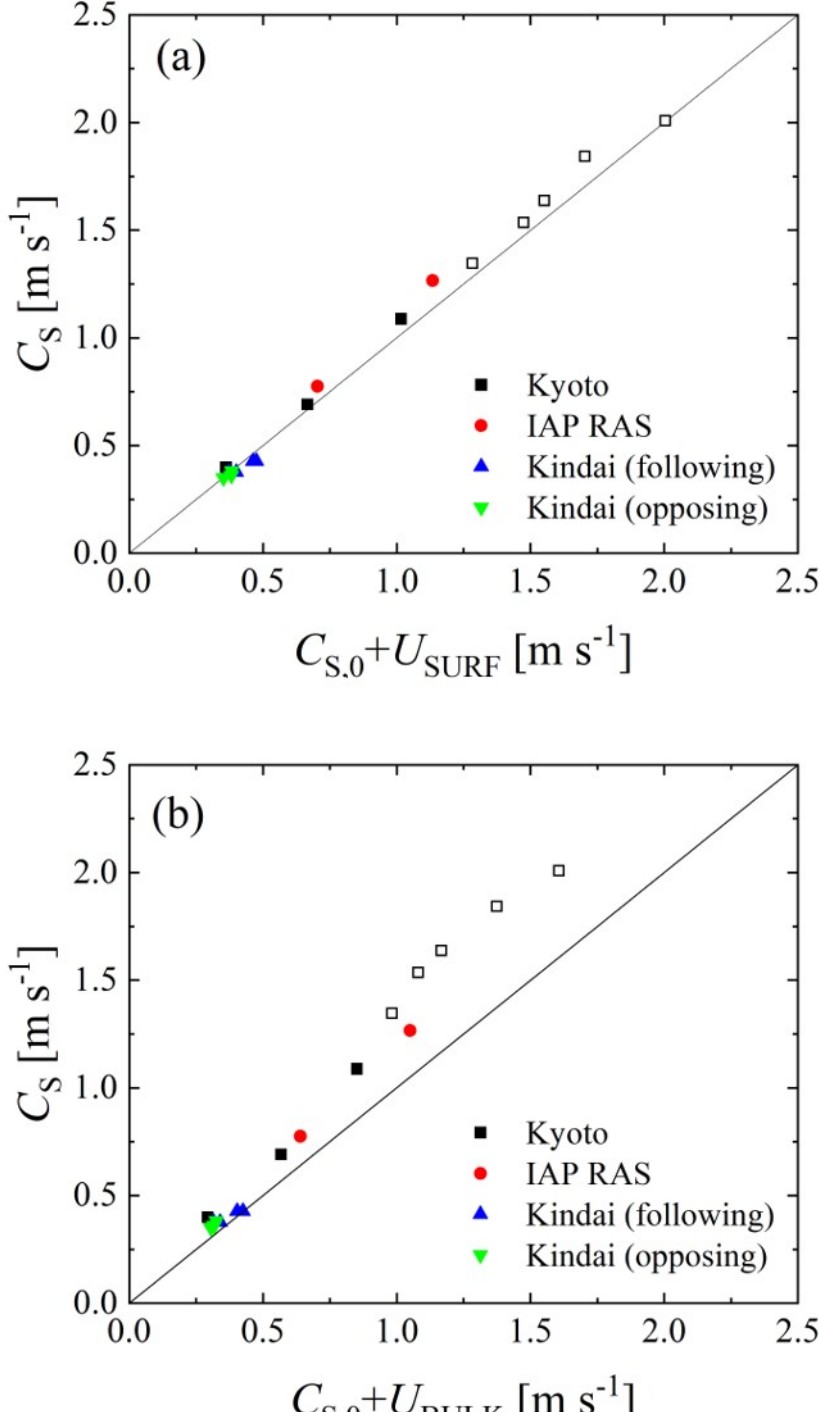

**Figure 6.** Relationship between phase velocity $C_S$ and (a) sum of $C_{S,0}$ and surface velocity of water $U_{SURF}$, and (b) sum of $C_{S,0}$ and bulk velocity of water $U_{BULK}$. Open symbols show the high-wind-speed cases.

$$\frac{\partial u}{\partial t} + u\frac{\partial u}{\partial x} + w\frac{\partial u}{\partial z} + \frac{1}{\rho}\frac{\partial p}{\partial x} = 0, \tag{3}$$

$$\frac{\partial w}{\partial t} + u\frac{\partial w}{\partial x} + w\frac{\partial w}{\partial z} + \frac{1}{\rho}\frac{\partial p}{\partial z} = -g, \tag{}$$

and the condition of non-compressibility:

$$\frac{\partial u}{\partial x} + \frac{\partial w}{\partial z} = 0, \tag{4}$$

with the kinematical

$$\frac{\partial \eta}{\partial t} + u\frac{\partial \eta}{\partial x} = w\Big|_{z=\eta(x,t)} \tag{5}$$

and dynamical boundary conditions

$$p\Big|_{z=\eta(x,t)} = 0 \tag{6}$$

at the water surface. Here, $u$ and $w$ are the horizontal and vertical velocity components, $p$ is the water pressure, $x$ and $z$ are the horizontal and upward vertical coordinates, $g$ is the gravity acceleration, and $\rho$ is the water density. The boundary condition at the bottom of the channel is $w\Big|_{z=-D} = 0$. It should be noted that the water depth in almost all the experimental runs exceeded half of the wavelength of the dominant waves (see Table 1). In this case, the deep-water approximation is applicable for describing the surface waves, and the boundary condition of the wave field vanishing with the distance from the water surface can also be used.

Because the fluid motion under consideration is 2D, the stream function can be introduced as follows:

$$u = \frac{\partial \psi}{\partial z}; w = -\frac{\partial \psi}{\partial x}. \tag{7}$$

To derive the linear dispersion relation for the surface waves at the plane shear flow with the horizontal velocity profile $U_\mathrm{w}(z)$, we consider the solution to Eqs. (3, 4) in terms of the stream function as the sum of the undisturbed state with steady shear flow and small-amplitude disturbances. Then, the stream function $\psi$ and pressure $p$ are as follows:

$$\psi(x,z,t) = \int^{z} U_w(z_1)dz_1 + \varepsilon\psi_1(x,z,t); \tag{8}$$

$$p(x,z,t) = -\rho gz + \varepsilon p_1(x,z,t), \tag{9}$$

where $\varepsilon \ll 1$, and the water elevation value is also the order of $\varepsilon$, namely $\varepsilon\eta_1(x, t)$.
In the linear approximation in $\varepsilon$, the system of Eqs. (3, 4) and the boundary
conditions of Eqs. (5, 6) take the form:

$$\left(\frac{\partial}{\partial t} + \frac{U_w(z)\partial}{\partial x}\right)\left(\frac{\partial^2 \psi_1}{\partial x^2} + \frac{\partial^2 \psi_1}{\partial z^2}\right) - \frac{\partial \psi_1}{\partial x}\frac{d^2 U_w(z)}{dz^2} = 0,$$


$$\frac{\partial \eta_1}{\partial t} + U_w(0)\frac{\partial \eta_1}{\partial x} = -\frac{\partial \psi_1}{\partial x}\bigg|_{z=0},$$

(10)

$$\frac{\partial p_1}{\partial x}\bigg|_{z=0} - \rho g\frac{\partial \eta_1}{\partial x} = 0,$$


$$\psi_1\bigg|_{z=-D} = 0.$$


Excluding $p_1$ with use of the first equation of the system in Eq. (3) and eliminating $\eta_1$
yields one boundary condition at the water surface for $\psi_1$:

$$\left[\left(\frac{\partial}{\partial t} + \frac{U_w(0)\partial}{\partial x}\right)^2\frac{\partial \psi_1}{\partial z} - \left(\frac{\partial}{\partial t} + U_w(0)\frac{\partial}{\partial x}\right)\frac{\partial \psi_1}{\partial x}\frac{dU_w}{dz} - g\frac{\partial^2 \psi_1}{\partial x^2}\right]\bigg|_{z=0} = 0.$$

(11)

For the harmonic wave disturbance, where

$$\psi_1(x, z, t) = \Psi(t)\exp(-i(\omega t - kt)),$$

(12)

substituting into Eqs. (10, 11) yields the Rayleigh equation for the complex amplitude of
the stream function disturbance:

$$(\omega - U_w(z)k)\left(\frac{d^2 \Psi_1}{dz^2} - k^2\Psi_1\right) + \frac{d^2 U_w(z)}{dz^2}k^2\Psi_1 = 0,$$

(13)

with the following boundary condition:

$$(\omega - U_w(0)k)^2\frac{d\Psi_1(0)}{dz} + (\omega - U_w(0)k)k\Psi_1(0)\frac{dU_w(0)}{dz} - k^2 g\Psi_1(0) = 0,$$

(14)

$$\Psi_1\bigg|_{z\to-\infty} \to 0.$$


Numerically solving the boundary layer problem for Eq. (13) with the boundary
conditions in Eq. (14) enables one to obtain the dispersion relation $\omega(k)$ for the surface
waves at the inhomogeneous shear flow. Note that because the phase velocity of the
waves significantly exceeded the flow velocity in all experiments (cf. Figs. 2 and 3), the
Rayleigh equation did not have a singularity, and the calculated frequency and phase
velocity of the wave were real values, i.e., the current was neutral stable.
The wave phase velocities $C_{\text{S-theor-l}} = \omega(k)/k$ were calculated for the parameters of
those experiments that contained complete information about the course and
characteristics of the waves, namely 1, 3, 5, 7, 9, 11, 13–15, 18, and 21–27 from Table 1.
The results are presented in Fig. 7a as the measured phase velocity $C_S$ versus calculated

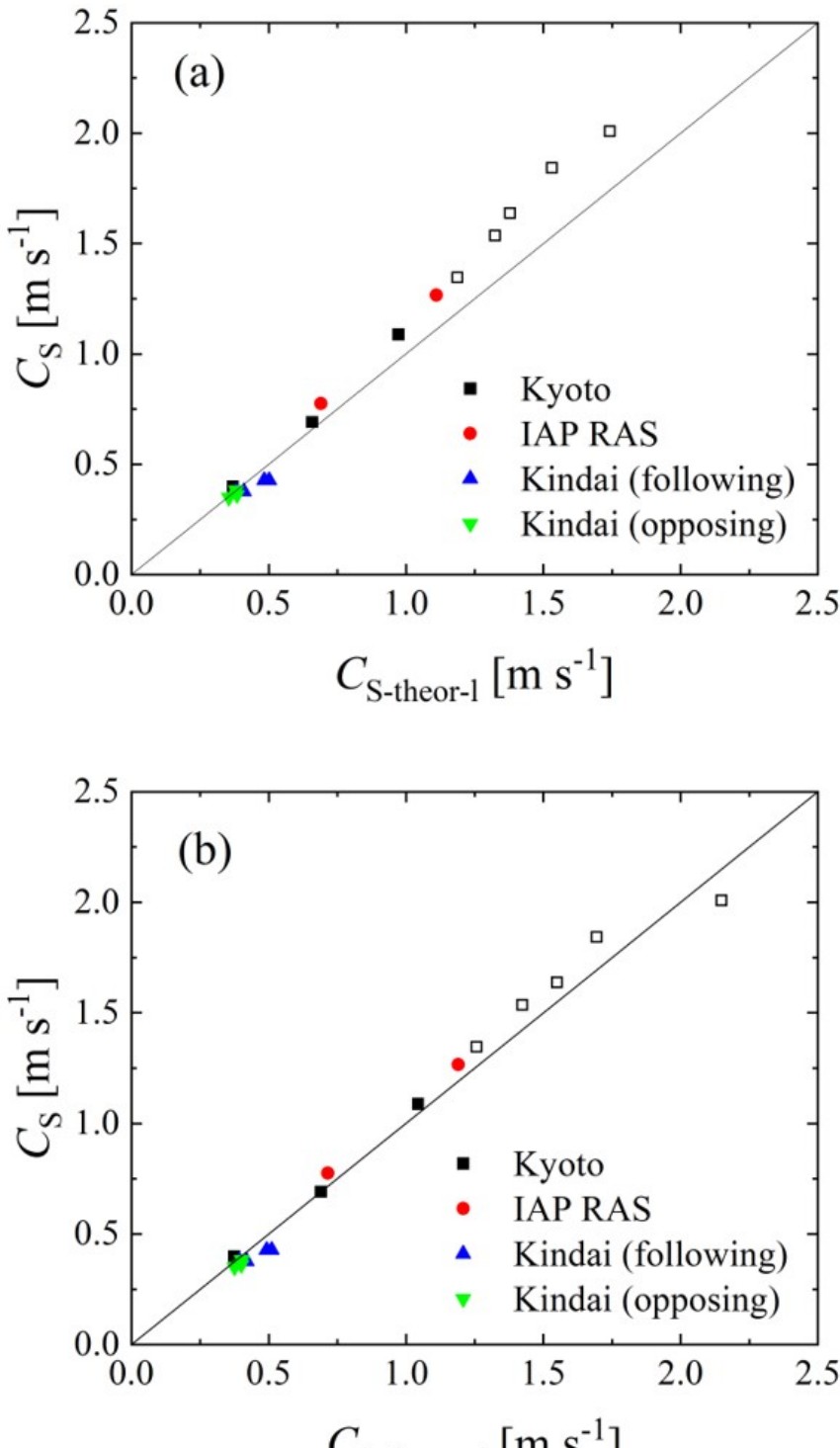


**Figure 7.** The measured phase velocity $C_S$ versus theoretical prediction: (a) linear model, and (b)

nonlinear model.



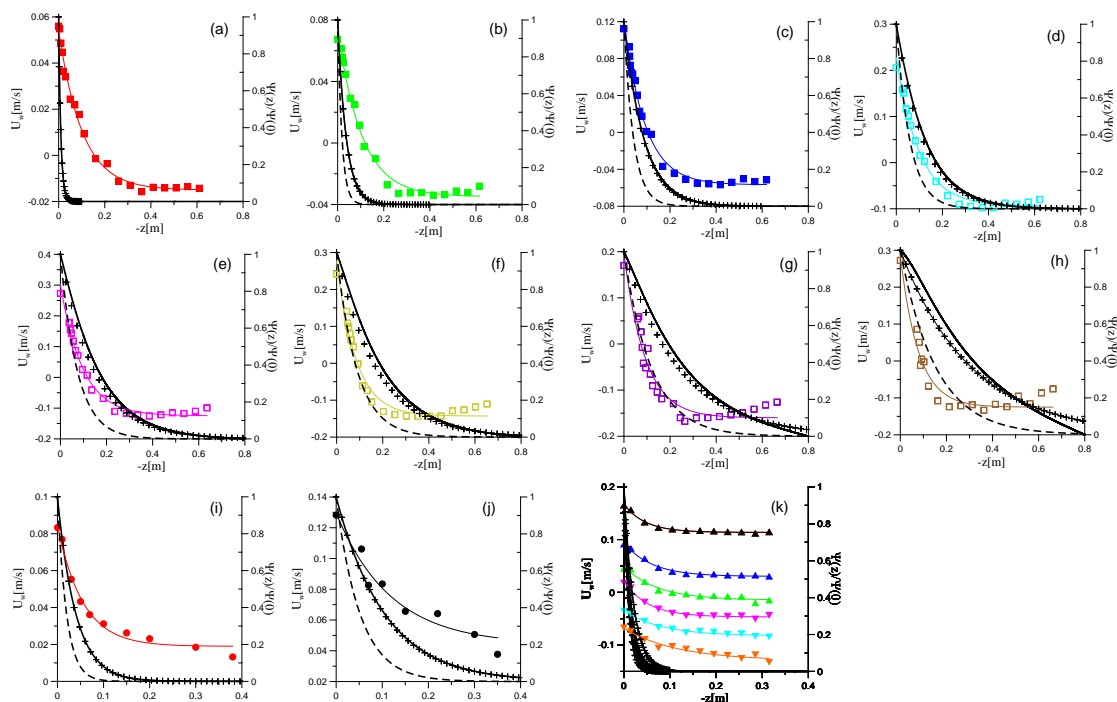

**Figure 8.** Vertical velocity profiles (points), their fitting (thin color line), the eigenfunction of the Eq. (8) with the boundary conditions Eq. (9) (black solid curve), the function $e^{kz}$ (crosses), the function $e^{2kz}$ (dashed line). The panels (a)-(j) corresponds to the experiments No. 1, 3, 5, 7, 9, 11, 13-15, 18 respectively, the panel (k) corresponds to the experiments No. 21-27.

phase velocity $C_{\text{S-theor-l}}$. One can see that the model corresponds to the data substantially better than does the model of linear potential waves at the homogeneous current $U_{\text{BULK}}$ (compare Fig. 6b). Considering the structure of the wave disturbances of the stream function, $\Psi_1(z)$, which was found as the eigenfunction of the boundary problem of Eqs. (11, 12). The profiles of $\Psi_1(z)$ are presented in Fig. 8. One can see that in all cases the functions $\Psi_1(z)$ are close to $e^{kz}$ at the background of the mean velocity profiles. Moreover, for experiments No. 1, 3, 5, 15, and 21–27 (see Fig. 8a, 8b, 8c, 8i, and 8k), the wave field is concentrated near the surface at a distance less than the scale of the change in the mean flow, where the flow velocity is approximately equal to $U_{\text{SURF}}$. This explains the good correlation in these cases of the observed phase velocity with the phase velocity of waves at the homogeneous current $U_{\text{SURF}}$ presented in Fig. 6a. At the same time, for experiments No. 7, 9, 5, 11, 13, 14, and 18 (see Figs. 8d–8h, and 8j), the scale of the variability of the flow is significantly smaller than the scale of the wave field. Under these conditions, a significant difference between the phase velocity of the waves and that given by the linear dispersion relation can be due to the influence of nonlinearity.

To estimate the nonlinear addition to the wave phase velocity, we used the results

of the weakly nonlinear theory of surface waves for the current with a constant shear. Of
course, the flow in the experiments of the present work does not have a constant shift, and
this was considered when obtaining the linear dispersion relation. However, it should be
taken into account that the contributions of the $n$-th harmonic to the nonlinear dispersion
relation are determined by wave fields in the $n$-power, which have a scale that is $n$ time
smaller than the first harmonic. Additionally, the model of constant shear of the mean
current velocity is already approximately applicable for the 2nd harmonic (see Fig. 8).
We use the nonlinear dispersion relation for waves in the current with a constant
shift in the deep-water approximation, which was obtained by Simmen and Saffman
(1985):

$$(\omega - U_w(0)k)^2 \frac{d\Psi_1(0)}{dz} + (\omega - U_w(0)k)k\Psi_1(0)\frac{dU_w(0)}{dz} - k^2 g\Psi_1(0) = \gamma(ka)^2,$$

$$\gamma = \frac{(\omega_0 - U_w(0)k)^2}{2k}\left(1 - \frac{1}{2}\Omega^2 + \left(1 + 2\Omega + \frac{1}{2}\Omega^2\right)^2\right),$$

$$\Omega = \frac{1}{(\omega_0 - U_w(0)k)}\frac{dU_w(0)}{dz},$$

(15)

Here, $\omega_0$ is the solution of the linear dispersion equation. Eq. (15) is rewritten in the
notation of this work and formulated in a reference frame in which the surface of the
water has the velocity $U_w(0)$. Note that the linear part of Eq. (15) coincides with Eq. (14).
The results of solving Eq. (15) are presented in Fig. 7b similarly to Fig. 7a as the
measured phase velocity $C_S$ versus calculated phase velocity $C_{S\text{-theor-nl}} = \omega(k)/k$, where
one can see their good agreement with each other. Thus, the wave frequency shift can be
explained by two factors, including the Doppler shift at the mean flow and the nonlinear
frequency shift, while, the latter can also be interpreted in its physical nature as the wave
frequency shift in the presence of its orbital velocities.
Recent studies have indicated a regime shift in the momentum, heat, and mass
transfer across an intensive broken wave surface along with the amount of dispersed
droplets and entrained bubbles at extreme high wind speeds over 30 m s$^{-1}$ (e.g., Powell et
al., 2003; Donelan et al., 2004; Takagaki et al., 2012, 2016; Troitskaya et al., 2012; Iwano
et al., 2013; Krall and Jähne, 2014; Komori et al., 2018; Krall et al., 2019). Thus, there is
the possibility of a similar regime shift in the Doppler shift of wind waves by the current
at extreme high wind speeds. However, the present study reveals that such a Doppler shift
is observed as under the conditions of normal wind speeds. In this case, the weakly
nonlinear approximation turns out to be applicable for describing the dispersion
properties of not only small-amplitude waves but also nonlinear and even breaking waves.
This implies that the intensive wave breaking at extreme high wind speeds occurs with
the saturation (or dumping) of the wave height rather than the wavelength. This evidence
might be helpful in investigating and modelling the wind-wave development at extreme
high wind speeds.
**4. Conclusion**
The effects of the current on wind waves were investigated through laboratory
experiments in three different wind-wave tanks with a pump at Kyoto University, Japan,
Kindai University, Japan, and IAP RAS. The study investigated 27 cases with the
measurements of winds, waves, and currents, at wind speeds ranging from 7–67 m s$^{-1}$. We
observed that the wind waves do not follow the dispersion relation in either the normal or
the extremely high wind speeds in the three tanks (Fig. 4)—excluding case 25, in which
the artificial current experiment used the Kindai tank. In case 25, $U_{\text{SURF}}$ is approximately
zero (Fig. 3); thus, the Doppler shift does not occur. Then, using 18 datasets (Kyoto and
IAP RAS tanks) (Fig. 5), we found that the ratio of $C_{\text{S}}/C_{\text{S},0}$ is constant at both normal and
extremely high wind speeds. Moreover, in the artificial current experiment in Kindai, we
observed that the ratio varies (Fig. 5). The evidence from the three tank experiments
implies that the same wave-current interaction occurs at normal and extremely high wind
speeds.
To develop an adequate model for wave-current interaction at normal and
extremely high wind speeds, we validated four models (Figs. 6 and 7). At normal wind
speeds under 30 m s$^{-1}$, the wave frequency, wavelength, phase velocity of waves, and
surface velocity of the water depended on the wind speed (Fig. 3). However, the bulk
velocity of the water showed a dependence on the tank type, i.e., a large tank with a
submerged wind-wave flume (IAP RAS) or wind flume above a tank (general type of
wind-wave tank) (Kyoto University) (Fig. 3). The effect of the Doppler shift was
confirmed at normal wind speeds, i.e., the significant waves were accelerated by the
surface flow, and the phase velocity was represented as the sum of the surface velocity of
water and the phase velocity, which is estimated by the dispersion relation of the
deep-water waves (Fig. 6). At extreme high wind speeds over 30 m s$^{-1}$, a Doppler shift
was observed similar to that under the conditions of normal wind speeds (Figs. 4 and 5).
This suggests that the Doppler shift is an adequate model for representing the acceleration
of wind waves by the current, not only for the wind waves at normal wind speeds but also
for those with intensive breaking at extreme high wind speeds. The data obtained by the
artificial current experiments conducted at Kindai University were used to explain how
the artificial current accelerates (or decelerates) the significant waves. A weakly
nonlinear model of surface waves at a shear flow was developed (Fig. 7). It was shown
that it describes well the dispersion properties of not only small-amplitude waves but also
strongly nonlinear and even breaking waves, typical for extreme wind conditions, with
speeds, $U_{10}$, exceeding 30 m s$^{-1}$.

**Data availability**
All analytical data used in this study are compiled in Table 1.

**Author contributions**
NT and NS planned the experiments, evaluated the data, and contributed equally to
writing the paper excluding Section 3.2. YT planed the Russia experiment, provided the
linear and non-linear models, prepared figures in Section 3.2, and contributed to writing
Section 3.2. CT prepared all figures excluding Section 3.2. NT performed the wind,
current, and wave measurements in the Kyoto experiment. NT, NS, and CT performed the
wind, current, and wave measurements in the Kindai experiment. AK and MV performed
the wind, current, and wave measurements in the Russia experiment.

**Competing interests**
The authors declare that they have no conflict of interest.

**Acknowledgements**
This work was supported by the Ministry of Education, Culture, Sports, Science and
Technology (Grant-in-Aid No's. 18H01284, 18K03953, and 19KK0087). This project
was supported by the Japan Society for the Promotion of Science and the Russian
Foundation for Basic Research (grant 18-55-50005, 19-05-00249, 20-05-00322) under
the Japan-Russia Research Cooperative Program. The experiments of IAP RAS were
partially supported by RSF (project 19-17-00209). We thank Prof. S. Komori and Mr.
Tsuji for their help in conducting the experiments and for useful discussions. The
experiments of IAP RAS were performed at the Unique Scientific Facility "Complex of
Large-Scale Geophysical Facilities" (http://www.ckp-rf.ru/usu/77738/).

**Appendix**
It is important to estimate the phase velocity and wavelength of the significant
wind-waves using the water-level fluctuation data. Here, we explain the method, called as
the cross-spectrum method. The water-level fluctuation $\eta(x, t)$ at arbitral location $x$ and
time $t$ is shown as the equation:

$$\eta(x,t) = \int_{-\Omega}^{\Omega} A(\omega) e^{i(\omega t - k(\omega)x)} d\omega \tag{A1}$$

where $\omega$ is the angular frequency, $A(\omega)$ is the complex amplitude, and $k(\omega)$ is the wavenumber of waves having $\omega$, $\Omega$ is the maximum angular frequency of the surface waves. $F_\eta(\omega)$ is the Fourier transformation of $\eta(x, t)$ when the measurement time $t_m$ and $\Omega$ are sufficiently large. Using the inverse Fourier transformation of $F_\eta(\omega)$, $\eta(x, t)$ is shown as:

$$\eta(x,t) = \frac{1}{2\pi} \int_{-\Omega}^{\Omega} F_\eta(\omega) e^{i\omega t} d\omega. \tag{A2}$$

Comparing Eqs. (A1, A2), $F_\eta(\omega)$ is $F_\eta(\omega) = 2\pi A(\omega) e^{-ik(\omega)x}$. Assuming that the wind waves change the shape little between two wave probes set upstream and downstream, we can set the upstream and downstream water-level fluctuations $\eta_1(t) = \eta(0, t)$ and $\eta_2(t) = \eta(\Delta x, t)$, respectively, with $\Delta x$ downstream from the first probe. The Fourier transformations $F_{\eta1}(\omega)$ and $F_{\eta2}(\omega)$ for $\eta_1(t)$ and $\eta_2(t)$, respectively, are shown as:

$$F_{\eta_1}(\omega) = 2\pi A(\omega), \tag{A3}$$

$$F_{\eta_2}(\omega) = 2\pi A(\omega) e^{-ik(\omega)\Delta x}. \tag{A4}$$

Then, the power spectra $S_{\eta1\eta1}(\omega)$ and $S_{\eta2\eta2}(\omega)$ for $\eta_1(t)$ and $\eta_2(t)$, respectively, are shown as:

$$S_{\eta_1\eta_1}(\omega) = \frac{1}{t_m} F_{\eta_1}^*(\omega) F_{\eta_1}(\omega) = \frac{1}{t_m} 4\pi^2 |A(\omega)|^2, \tag{A5}$$

$$S_{\eta_2\eta_2}(\omega) = \frac{1}{t_m} F_{\eta_2}^*(\omega) F_{\eta_2}(\omega) = S_{\eta_1\eta_1}(\omega). \tag{A6}$$

Here, the superscript * indicates the complex conjugate number. The cross-spectrum $Cr(\omega)$ for $\eta_1(t)$ and $\eta_2(t)$ is shown as:

$$Cr(\omega) = \frac{1}{t_m} F_{\eta1}^*(\omega) F_{\eta2}(\omega) = \frac{1}{t_m} 4\pi^2 |A(\omega)|^2 e^{ik(\omega)\Delta x}. \tag{A7}$$

Using Euler's theorem, Eq. (A7) transforms to:

$$Cr(\omega) = \frac{1}{t_m} 4\pi^2 |A(\omega)|^2 (\cos k(\omega)\Delta x + i \sin k(\omega)\Delta x)$$

$$= S_{\eta_1}(\omega)(\cos k(\omega)\Delta x + i \sin k(\omega)\Delta x). \tag{A8}$$

The cospectrum $Co(\omega)$ and quad spectrum $Q(\omega)$ are defined as the real and imaginary parts of $Cr(\omega)$, respectively, shown as $Cr(\omega) = Co(\omega) + iQ(\omega)$. Moreover, the phase $\theta(\omega)$ is defined as $\theta(\omega) = \tan^{-1}(Q(\omega)/Co(\omega))$. Thus, $\theta(\omega)$ can be calculated as:

$$\theta(\omega) = \tan^{-1}(\tan(k(\omega)\Delta x) = k(\omega)\Delta x. \tag{A9}$$

Generally, the velocity of the wind waves $C$ is defined as:

$$C = \frac{\omega}{k} = \frac{L}{T}, \tag{A10}$$


where $L$ is the wavelength and $T$ is the wave period. From Eqs. (A9, A10), $C(\omega)$ and $L(\omega)$
can be transformed to

$$C(\omega) = \frac{\omega}{k} = \frac{\omega \Delta x}{\theta(\omega)}, \tag{A11}$$


$$L(\omega) = \frac{2\pi}{k} = \frac{2\pi \Delta x}{\theta(\omega)}. \tag{A12}$$


When we estimate the phase $\theta_m(\omega_m)$ at the angular frequency of significant wind-waves
$\omega_m (=2\pi f_m)$, the phase velocity of the significant wind waves $C_S (= C(\omega_m))$ and significant
wavelength $L_S (= L(\omega_m))$ are calculated by:

$$C_S = \frac{2\pi f_m \Delta x}{\theta(f_m)}, \tag{A13}$$


$$L_S = \frac{2\pi \Delta x}{\theta(f_m)}. \tag{A14}$$


In the study, $C_S$ and $L_S$ are estimated by Eqs. (A13, A14) using the cross-spectrum
method.

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
