# Peer review of "Effects of current on wind waves in strong winds"

_Ocean Science, 2020_

## Referee Comment (RC1) · Anonymous Referee #1 · 27 May 2020

This is a nice paper, and the agreement between the observations and the nonlinear dispersion relationship theory was particularly satisfying. I do have a number of suggestions and comments below, which I would like to see addressed to strengthen the paper further.

1. Line 26 (and 357): Statement "The results show that 27 different types of currents" is confusing, as it implies there are 27 categories of currents this study uncovered. I am guessing authors meant that this study investigated 27 cases with carrying winds, waves, and currents.

2. Line 44-45: maximum ocean surface current velocity is certainly more than 1 m/s.

3. Introduction: Other than the high winds and the lack of data, the introduction seems

to be missing a scientific objective or a hypothesis. Please elaborate what reasons did the authors have to doubt that waves might not follow the dispersion relationship in high winds. Why wouldn't they?

4. Table: How were U10 and U* calculated from the wind speed measurements in the tunnel? This question becomes especially intriguing as wind speed studied here goes far beyond applicability limits of any Cd parameterization.

5. Table: What is freestream wind velocity? How is it defined and calculated?

6. Methodology: Paper's conclusions could have been reached based on a single tank experiment. Why use three tanks? Is that because no single tank had all required capabilities (e.g., high wind vs current control, etc.)? Please add an explanation in the methodology section.

7. Lines 98-99 How was the phase speed Cs calculated? Lines 98-99 mentioned some "cospectra method" and refer to Takagaki et al., 2017, but I looked through that article and did not find it. I think this method should be presented in greater detail in this manuscript. It is important to understand if the underlying currents, including the surface drift current, as well as the observed sharp vertical profile, might skew this estimate.

8. Line 133: What is an "open tank"? Please explain in the manuscript.

---

## Author Comment (AC1) · 25 Jun 2020

Reviewer #1:

We would like to thank Reviewer #1 for carefully reviewing our manuscript and providing invaluable advice.

Please find PDFs for reply to reviewer 1 and supplement for revision tracking.

Sincerely yours, Naohisa Takagaki

Please also note the supplement to this comment:
https://os.copernicus.org/preprints/os-2020-30/os-2020-30-AC1-supplement.zip

---

## Referee Comment (RC2) · Anonymous Referee #1 · 9 Jul 2020

All of my comments were adequately addressed. I recommend the revised article for publication in its present form.
* * *

---

## Referee Comment (RC3) · Anonymous Referee #2 · 20 Jul 2020

This paper reports results of the laboratory investigations of the current-induced Doppler shift in wind waves at moderate to extremely high wind speeds. Experiments performed at three different tanks: Kyoto University (Japan), Kindai University (Japan), and the Institute of Applied Physics (Russia). Profiles of the wind and current velocities, and the surface wave papameters (namely phase velocity of dominant wave) were measured at all the facilities.

Results of the measurement in three different tanks are consistent provide description of the Doppler shift in the wind speeds range from 7 to 67 m/s. As found, phase velocity of dominant waves can be represented as the sum of the surface current and phase velocity estimated through the dispersion relation. The authors showed that such model is valid either for moderate and extremely high winds, even if the dominant

waves are subjected to intensive breaking. To interpret the measurements, a weakly nonlinear model of surface waves at a shear flow is suggested.

The paper is well and clear written, it contains very interesting experimental material which after some efforts, can be applied for the open ocean conditions. I recommend to publish this paper as it is.

—————————————————————

---

## Author Comment (AC2) · 22 Jul 2020

Reviewer #1:

We would like to thank reviewer #1 for reading our manuscript carefully and giving us the positive response. We start to prepare the revised manuscript.

Sincerely yours, Naohisa Takagaki

———————————————————

---

## Author Comment (AC3) · 22 Jul 2020

Reviewer #2:

We would like to thank reviewer #2 for reading our manuscript carefully and giving us the favorable reply.

Sincerely yours, Naohisa Takagaki
* * *

---

## Author Response (AR1)

**Reviewer #1:**

**Reply: We revised the manuscript according to your reviewing. Following are the point-by-point response to reviewer #1 comments for specifying all changes in the mark-up manuscript.**

*1. Line 26 (and 357): Statement "The results show that 27 different types of currents" is confusing, as it implies there are 27 categories of currents this study uncovered. I am guessing authors meant that this study investigated 27 cases with carrying winds, waves, and currents.*

**Previous Author's Reply: We revised the sentences in lines 26 and 357 in the manuscript.**

**Change in Manuscript: We revised the sentences in lines 26-28 in the mark-up manuscript. We revised the sentences in lines 398-400 in the mark-up manuscript.**

*2. Line 44-45: maximum ocean surface current velocity is certainly more than 1 m/s.*

**Previous Author's Reply: We revised the sentence in line 46 in the manuscript.**

**Change in Manuscript: We revised the sentences in lines 46-47 in the mark-up manuscript.**

*3. Introduction: Other than the high winds and the lack of data, the introduction seems to be missing a scientific objective or a hypothesis. Please elaborate what reasons did the authors have to doubt that waves might not follow the dispersion relationship in high winds. Why wouldn't they?*

**Previous Author's Reply: Wind waves follow the dispersion relationship at normal wind speeds. At extremely high wind speeds, the water surface is intensively broken because of strong surface wind shear (e.g. Donelan et al., 2004; Troitskaya et al., 2012, 2017, 2018a, 2018b; Takagaki et al., 2012, 2016; Holthuijsen et al., 2012). Thus, it is unclear if such wind waves with the surface foam layer at extremely high wind speeds have properties similar to those of the wind waves at normal wind speeds. We added a detailed explanation to the introduction of the revised manuscript.**

**Change in Manuscript: We revised the sentences in lines 54-70 in the mark-up manuscript.**

*4. Table: How were U10 and U\* calculated from the wind speed measurements in the tunnel? This question becomes especially intriguing as wind speed studied here goes far beyond applicability limits of any Cd parameterization.*

**Previous Author's Reply: The values of $U_{10}$ and $u$\* taken in Kyoto University (Cases 1-14) were estimated by the eddy correlation method using a laser Doppler anemometer (LDA) at normal wind speeds and a phase Doppler anemometer (PDA) at high wind speeds. The details were written in previous papers by the authors (e.g. Takagaki et al., 2012; Iwano et al., 2013; Komori et al., 2018), where we briefly explain the method: the air velocity and Reynolds stress were measured at a fetch of $x$ = 6.5 m using a phase Doppler anemometer (Dantec Dynamics PDA). Laser beams were shot through the plate-glass sidewall, and we prepared a small droplet-adherent prevention device (DAPD) to avoid irregular reflection by the wall and impingement of the droplets dispersed from the intensively breaking wind waves. The DAPD had a size of 0.07 m x 0.07 m x 0.007 m and was fixed on the inside glass wall (Fig. A1). Four orifices with a diameter of 0.005 m were installed on the device, and four laser beams were introduced through the orifices into the test section. Clean, compressed air was blown through the orifices along the plate-glass sidewall. Therefore, even if dispersed droplets impinged on and adhered to the orifices, the compressed air removed the droplets, creating a clear path for the laser beam. As the PDA enabled us to simultaneously measure the diameter ($d_p$), streamwise and vertical velocities, and number of dispersed droplets, we could measure the streamwise air velocity ($U_p$) and the product of the streamwise and vertical air velocity fluctuations ($u_p v_p$) by conditionally extracting the signals from dispersed droplets of $d_p$ < 30 μm. Thus, we defined the mean velocity ($U$) and the Reynolds stress ($-uv$) in air as the ensemble averaged values ($$ and $< u_p v_p >$) of $U_p$ and $u_p v_p$ for droplets of $d_p$ < 30 μm. The air friction velocity ($u$\*) was estimated by an eddy correlation method as $u* = \left(-\langle u_p v_p \rangle\right)^{1/2}$, because the shear stress at the interface ($\tau$) was defined by $\tau = \rho u^{*2} = \rho C_D U_{10}^2$. The value of $\left(-\langle u_p v_p \rangle\right)^{1/2}$ was estimated by extrapolating the measured values of the Reynolds stress to the mean surface $z$ = 0 m. The $U_{10}$ was estimated by the log-law: $U_{10}$ - $U_{min}$ = $u$\*/$\kappa$ln($z_{10}$/$z_{min}$), where $U_{min}$ is the air velocity nearest the water surface ($z_{min}$), and $z_{10}$ is 10 m. Moreover, the drag coefficient $C_D$ was estimated by $C_D = (u*/U_{10})^2$.**

   **The values of $U_{10}$ and $u$\* taken in Kindai University (Cases 21-27) were estimated by the empirical curve by Iwano et al. (2013), which was proposed by the above eddy correlation method.**

   **The values of $U_{10}$ and $u$\* taken in IAP RAS (Cases 15-20) were taken from Troitskaya**

et al. (2012) by a Pitot tube, where $u^*$ was estimated by a profile method considering the profiles in the constant flux layer and the wake region:

$$U_\infty - U(z) = u^* \left( -\frac{1}{\kappa} ln(z/\delta) + \alpha \right); \ z/\delta < 0.15, \quad (A1)$$

$$U_\infty - U(z) = \beta u^* (1 - z/\delta)^2; \ z/\delta > 0.15, \quad (A2)$$

respectively. Here, $U_\infty$ is the freestream wind speed, $\delta$ is the boundary layer thickness; $\alpha$ and $\beta$ are the constant values that depend on flow fields. The constant values of $\alpha$ and $\beta$ are calibrated at low wind speeds without the dispersed droplets. Measuring the profile in the constant flux layer (Eq. A1) in extremely high wind speeds is difficult because of the large breaking waves and dispersed droplets. Thus, using $\beta$ measured at low wind speeds, $u^*$ is estimated by Eq. (A2) at extremely high wind speeds. The value of $U_{10}$ is estimated by Eq. (A1) at $z = 10$ m with $\alpha$ measured at normal wind speeds. The value of $C_D$ is estimated by $\tau = \rho u^{*2} = \rho C_D U_{10}^2$. Although the measurement methods for $u^*$, $U_{10}$, and $C_D$ in IAP RAS and Kyoto are different, the values approximately correspond to each other (see Troitskaya et al. (2012) and Takagaki et al. (2012)).

We added these methods for estimating $u^*$, $U_{10}$, and $C_D$ in the revised manuscript.

Change in Manuscript: We revised the sentences in lines 104-112, lines 123-136, and lines 151-153 in the mark-up manuscript.

[Figure]

**Fig. A1: Droplet-adherent prevention device attached to the plate-grass sidewall. (a) View on the *x*–*z* plane and (b) view on the *y*–*z* plane; *x*, *y*, and *z* represent the streamwise, spanwise, and vertical directions, respectively. The four circles in (a) indicate the four orifices.**

*5. Table: What is freestream wind velocity? How is it defined and calculated?*

**Previous Author's Reply: The freestream wind velocity is the wind velocity in the freestream region, that is, the wind velocity outside the boundary layer, which is the constant velocity. The freestream wind speed is calculated as the average velocity in the freestream region.**

**Change in Manuscript: We didn't revise any sentences in the mark-up manuscript.**

*6. Methodology: Paper's conclusions could have been reached based on a single tank experiment. Why use three tanks? Is that because no single tank had all required capabilities (e.g., high wind vs current control, etc.)? Please add an explanation in the methodology section.*

**Previous Author's Reply: We obtained the conclusion through experiments using three tanks. We observed that wind waves do not follow the dispersion relation at either normal or the**

extremely high wind speeds in the three tanks—excluding case 25, which was the artificial current experiment using the Kindai tank (Fig. 4). In case 25, $U_{\text{SURF}}$ is approximately zero; thus, the Doppler shift does not occur in this situation, and the results in Fig. 4 were obtained from the three tanks. Then, using 18 datasets (Kyoto and IAP RAS tanks), we found that the ratio of $C_S/C_{S,0}$ is constant at normal and extremely high wind speeds (Fig. 5), implying that the same wave-current interaction occurs at normal and extremely high wind speeds. From the artificial current experiment in Kindai, we observed that the ratio varies (Fig. 5). Thus, the results in Fig. 5 were also obtained with the three tanks. We used 17 datasets—which included current profiles—to investigate the empirical and theoretical model at both normal and extremely high wind speeds (Figs. 6 and 7). Because the explanation in the previous manuscript may mislead readers, we added this detailed explanation, which is equivalent to the methodology, to the conclusion section of the revised manuscript.

**Change in Manuscript: We revised the sentences in lines 401-411 in the mark-up manuscript.**

*7. Lines 98-99 How was the phase speed Cs calculated? Lines 98-99 mentioned some "cospectra method" and refer to Takagaki et al., 2017, but I looked through that article and did not find it. I think this method should be presented in greater detail in this manuscript. It is important to understand if the underlying currents, including the surface drift current, as well as the observed sharp vertical profile, might skew this estimate.*

**Previous Author's Reply: We agree with your statement, changed the name to "cross-spectrum method", and added the explanation in an appendix of the revised manuscript for calculating the phase velocity $C_S$ and wavelength $L_S$. The following is the explanation.**

**Cross-spectrum method**

The water-level fluctuation $\eta(x, t)$ at an arbitrary location $x$ and time $t$ is shown as the equation:

$$\eta(x,t) = \int_{-\Omega}^{\Omega} A(\omega)e^{i(\omega t - k(\omega)x)}d\omega \tag{A1}$$

where $\omega$ is the angular frequency, $A(\omega)$ is the complex amplitude, and $k(\omega)$ is the wavenumber of waves having $\omega$, $\Omega$ is the maximum angular frequency of surface waves. The $F_\eta(\omega)$ is the Fourier transformation of $\eta(x, t)$ when the measurement time $t_m$ and $\Omega$ are sufficiently large. Using the inverse Fourier transformation of $F_\eta(\omega)$, $\eta(x, t)$ is shown as:

$$\eta(x,t) \;=\; \frac{1}{2\pi}\int_{-\Omega}^{\Omega} F_\eta(\omega)e^{i\omega t}\,d\omega. \tag{A2}$$

Comparing Eq. (A1) and (A2), $F_\eta(\omega)$ is $F_\eta(\omega) = 2\pi A(\omega)e^{-ik(\omega)x}$. Assuming that the wind waves change the shape little between two wave probes set upstream and downstream, we set the upstream and downstream water-level fluctuations to $\eta_1(t) = \eta(0,\,t)$ and $\eta_2(t) = \eta(\Delta x,\,t)$, respectively, with $\Delta x$ downstream from the first probe. The Fourier transformations $F_{\eta1}(\omega)$ and $F_{\eta2}(\omega)$ for $\eta_1(t)$ and $\eta_2(t)$, respectively, are shown as:

$$F_{\eta_1}(\omega) \;=\; 2\pi A(\omega), \tag{A3}$$

$$F_{\eta_2}(\omega) \;=\; 2\pi A(\omega)e^{-ik(\omega)\Delta x}. \tag{A4}$$

Then, the power spectra $S_{\eta1\eta1}(\omega)$ and $S_{\eta2\eta2}(\omega)$ for $\eta_1(t)$ and $\eta_2(t)$, respectively, are shown as:

$$S_{\eta_1\eta_1}(\omega) = \frac{1}{t_{\mathrm m}}F_{\eta_1}^{*}(\omega)F_{\eta_1}(\omega) = \frac{1}{t_{\mathrm m}}4\pi^2|A(\omega)|^2, \tag{A5}$$

$$S_{\eta_2\eta_2}(\omega) = \frac{1}{t_{\mathrm m}}F_{\eta_2}^{*}(\omega)F_{\eta_2}(\omega) = S_{\eta_1\eta_1}(\omega). \tag{A6}$$

Here, the superscript * indicates the complex conjugate number. The cross-spectrum $Cr(\omega)$ for $\eta_1(t)$ and $\eta_2(t)$ is shown as:

$$Cr(\omega) = \frac{1}{t_m}F_{\eta1}^{*}(\omega)F_{\eta2}(\omega) = \frac{1}{t_m}4\pi^2|A(\omega)|^2 e^{ik(\omega)\Delta x}. \tag{A7}$$

Using Eular's theorem, Eq. (A7) transforms to:

$$Cr(\omega) = \frac{1}{t_{\mathrm m}}4\pi^2|A(\omega)|^2(\cos k(\omega)\Delta x + i\sin k(\omega)\Delta x)$$

$$= S_{\eta_1}(\omega)(\cos k(\omega)\Delta x + i\sin k(\omega)\Delta x). \tag{A8}$$

We can define the cospectrum $Co(\omega)$ and quad spectrum $Q(\omega)$ as the real and imaginary parts of Cr($\omega$), respectively, shown as:

$$Cr(\omega) = Co(\omega) + iQ(\omega) \tag{A9}$$

Moreover, the phase $\theta(\omega)$ is defined as:

$$\theta(\omega) = \tan^{-1}\left(\frac{Q(\omega)}{Co(\omega)}\right), \tag{A10}$$

thus,

$$\theta(\omega) = \tan^{-1}(\tan(k(\omega)\Delta x) = k(\omega)\Delta x. \tag{A11}$$

Generally, the velocity of the wind waves $C$ is defined as:

$$C = \frac{\omega}{k} = \frac{L}{T}, \tag{A12}$$

where $L$ is the wavelength and $T$ is the wave period. From Eqs. (A11) and (A12), $C(\omega)$ and $L(\omega)$ can be transformed to:

$$C(\omega) = \frac{\omega}{k} = \frac{\omega \Delta x}{\theta(\omega)}, \tag{A13}$$

$$L(\omega) = \frac{2\pi}{k} = \frac{2\pi \Delta x}{\theta(\omega)}. \tag{A14}$$

When we estimate $\theta_m(\omega_m)$ at the significant angular frequency of wind waves $\omega_m$ ($=2\pi f_m$), the phase velocity of the significant wind waves $C_S(\omega_m)$ and significant wavelength $L_S(\omega_m)$ are calculated by:

$$C_S = \frac{2\pi f_m \Delta x}{\theta(f_m)}, \tag{A15}$$

$$L_S = \frac{2\pi \Delta x}{\theta(f_m)}. \tag{A16}$$

**Change in Manuscript: We revised the sentences in line 119, lines 138-142, lines 156-157, and lines 462-507 (appendix) in the mark-up manuscript.**

*8. Line 133: What is an "open tank"? Please explain in the manuscript.*

**Previous Author's Reply: The typhoon simulation tank in IAP RAS is constructed with a large tank and a submerged wind-wave flume. The operating cross section of the airflow is 0.40 x 0.40 m², and the sidewalls are submerged at a depth of 0.30 m (see Troitskaya et al., 2012). We removed the term "open tank" in the revised manuscript and changed the sentence to "This is because the wind-wave flume at IAP RAS is a submerged flume; thus, the Stokes drift on the wavy water surface does not provide the counterflow for the bulk water, unlike in the closed tank at Kyoto University" in the revised manuscript.**

**Change in Manuscript: We revised the sentences in lines 171-175, lines 414-416 in the mark-up manuscript.**

**Reviewer #2:**

Reply: We would like to thank reviewer #2 for reading our manuscript carefully and giving us the favorable reply. We revised the manuscript according to all reviewers. Please fined the mark-up manuscript.

[revised manuscript text omitted]
_{\text{m}}$, $U_{\text{SURF}}$, and $U_{\text{BULK}}$ vary, $L_{\text{S}}$ and $C_{\text{S}}$

are concentrated at single points at $L_{\text{S}} = 0.1$ m and $C_{\text{S}} = 0.4$ m s$^{-1}$, respectively. This shows that the intensity and direction of the current do not significantly affect $L_{\text{S}}$ and $C_{\text{S}}$ but do affect $f_{\text{m}}$ and $U_{\text{SURF}}$. Thus, this implies that the present artificial current changes the water flow dramatically but does not affect the development of the wind waves.

Figure 4 shows the dispersion relation and demonstrates that the Kindai data points depend on the variation in the water velocity of the artificial current. The plots for the Kyoto University and IAP RAS cases at normal wind speeds (solid symbols) are concentrated above the solid curve, showing the dispersion relation of the deep-water waves ($\omega^2 = gk$). Meanwhile, the plots for extreme high wind speeds (open symbols) are also concentrated above the solid curve. This implies that the wind waves, along with the intensive breaking at extreme high wind speeds, are dependent on the Doppler shift. To investigate the phase velocity trend, Fig. 5 shows the ratio of the measured phase velocity

$C_{\text{S}}$ to the phase velocity $C_{\text{S},0}$ estimated by the dispersion relation of the deep-water waves

$(C_{\text{S},0} = (gL_{\text{S}}/2\pi)^{1/2})$ against the wind velocity. From the figure, the ratios at the normal wind speeds assume a constant value (~1.21 in Kyoto or ~1.27 in IAP RAS). Moreover, the ratios at the extreme high wind speeds take similar values of 1.23 and 1.28 for Kyoto or IAP RAS, respectively. This implies that the phase velocities at extreme high wind speeds are accelerated by the current just as those at normal wind speeds. However, the

Kindai values are scattered and increase in the following cases and decrease in the opposing cases. It is clear that the artificial current accelerates (or decelerates) the phase velocity.

To interpret the relationship among the measured phase velocity $C_{\text{S}}$, first phase velocity $C_{\text{S},0}$ estimated by the dispersion relation, and water velocity, two types of phase velocity were evaluated: the sum of $C_{\text{S},0}$ and surface velocity of water $U_{\text{SURF}}$ and the sum of $C_{\text{S},0}$ and bulk velocity of water $U_{\text{BULK}}$. Figure 6 shows the relationship between $C_{\text{S}}$ and (a) $C_{\text{S},0} + U_{\text{SURF}}$, and (b) $C_{\text{S},0} + U_{\text{
[revised manuscript text omitted]